# Transplantation of Human Induced Pluripotent Stem Cell-Derived Neural Progenitor Cells Promotes Forelimb Functional Recovery after Cervical Spinal Cord Injury

**DOI:** 10.3390/cells11172765

**Published:** 2022-09-05

**Authors:** Yiyan Zheng, Chrystine M. Gallegos, Haipeng Xue, Shenglan Li, Dong H. Kim, Hongxia Zhou, Xugang Xia, Ying Liu, Qilin Cao

**Affiliations:** 1Center for Translational Science, Florida International University, 11350 SW Village Pkwy, Port St. Lucie, FL 34987, USA; 2Robert Stempel College of Public Health and Social Work, Florida International University, 11350 SW Village Pkwy, Port St. Lucie, FL 34987, USA; 3Department of Neurosurgery, McGovern Medical School, University of Texas Health Science Center at Houston, 6431 Fannin St., Houston, TX 77030, USA; 4Center for Stem Cell and Regenerative Medicine, The Brown Foundation Institute of Molecular Medicine for the Prevention of Human Diseases, University of Texas Health Science Center at Houston, Houston, TX 77030, USA

**Keywords:** iPSC, spinal cord injury, neural repair, neuroprotection, transplantation

## Abstract

Locomotor function after spinal cord injury (SCI) is critical for assessing recovery. Currently, available means to improve locomotor function include surgery, physical therapy rehabilitation and exoskeleton. Stem cell therapy with neural progenitor cells (NPCs) transplantation is a promising reparative strategy. Along this line, patient-specific induced pluripotent stem cells (iPSCs) are a remarkable autologous cell source, which offer many advantages including: great potential to generate isografts avoiding immunosuppression; the availability of a variety of somatic cells without ethical controversy related to embryo use; and vast differentiation. In this current work, to realize the therapeutic potential of iPSC-NPCs for the treatment of SCI, we transplanted purified iPSCs-derived NPCs into a cervical contusion SCI rat model. Our results showed that the iPSC-NPCs were able to survive and differentiate into both neurons and astrocytes and, importantly, improve forelimb locomotor function as assessed by the grooming task and horizontal ladder test. Purified iPSC-NPCs represent a promising cell type that could be further tested and developed into a clinically useful cell source for targeted cell therapy for cervical SCI patients.

## 1. Introduction

Human induced pluripotent stem cells (hiPSCs), reprogrammed from somatic cells by overexpression of four transcription factors, share properties similar to human embryonic stem cells (ESCs) in nearly every aspect examined, including: differentiation potentials into all three germ layers; expression of pluripotency genes; methylation state; and the formation of teratomas in vivo [1,2,3]. Compared with ESCs, iPSCs offer significant additional advantages, such as the availability of a variety of somatic cells without ethical controversy related to embryo use and, in particular, the great potential to produce autografts without the need for immunosuppression. Thus, hiPSCs offer tremendous potential in precision medicine for personalized therapy [4,5,6,7,8]. However, its successful clinical translation is still lacking, although a couple of clinical trials are pending in Japan [9,10]. To realize the therapeutic potential of hiPSC-derived neural stem cells (NSCs) or neural progenitor cells (NPCs) for neurological diseases, including spinal cord injury (SCI), more preclinical studies are needed to examine the important aspects of stem cell treatment, including the survival and differentiation/integration of grafted cells, the therapeutic efficacy, and long-term safety.

SCI is a devastating neurological condition which results in significant physical disabilities due to the loss of both sensory and motor functions below the injury. Therapeutic interventions that restore even partial function would significantly increase the patients’ quality of life. Unfortunately, effective therapies, which could bring meaningful functional recovery after SCI, are currently unavailable. Human iPSC-based transplantation therapy could bring new hope to patients with SCI [11,12,13]. Indeed, hiPSC derived NSCs or NPCs have been shown to survive and differentiate into neurons and glial cells in the injured spinal cord after transplantation [14,15,16]. Interestingly, the NPC grafts have been shown to overcome the inhibitory environment to extend axons, which have the potential to subsequently form neural circuits with host axons [14,17,18,19], suggesting that grafted NPCs could provide neuronal relays reconnecting the injured spinal cord. Importantly, transplantation of hiPSC-derived NSCs or NPCs can promote functional recovery after SCI [20,21,22]. Most previous pre-clinical studies have primarily used thoracic SCI models focusing on hindlimb functional recovery. However, most clinical SCI cases occur at the cervical level [23]. Furthermore, recovery of hand and forearm function, which is one of the highest priorities to individuals with SCI, is best studied using cervical models.

In our present work, we have established a rat contusion model at cervical level 5 (C5) of the spinal cord, transplanted purified hiPSC-NPCs, evaluated anatomical integration, and assessed locomotor recovery with behavioral tests for upper limb movement. Our results showed that grafted hiPSC-NPCs were able to integrate with the host spinal cord and improve functional locomotor recovery as evidenced by grooming and horizontal ladder tests. Our results suggest that transplantation of hiPSC-derived NPCs is a potentially effective treatment for patients with cervical SCI. 

## 2. Materials and Methods

### 2.1. Culture of hiPSCs

Human iPSC Nestin-ZsGreen knockin reporter cell line, obtained from Dr. Mahendra Rao at Center for Regenerative Medicine, National Institutes of Health, were maintained in chemically defined mTeSR medium (Stem Cell Technologies, Vancouver, BC, Canada) in a feeder free fashion and passaged every 4–5 days at a 1:4~1:6 ratio using 0.5 mM EDTA following the manufacturer’s instructions, as described previously [24].

### 2.2. Differentiation of iPSCs into Neural Progenitor Cells

After being digested in 0.5 mM EDTA for 10 min at 37 °C, small clumps of human iPSCs were then transferred to Petri dishes (Corning, Corning, NY, USA) and cultured in iPSC medium without basic fibroblast growth factor (bFGF) for 8 days to form embryoid bodies (EBs) as previously described [25,26]. The EBs were then transferred to cell culture plates and continued to differentiate in neural induction medium (DMEM/F12 plus Glutamax, 1× non-essential amino acid (NEAA), 1× N2 supplement, and bFGF (20 ng/mL)). After differentiation for 2–3 days, neural rosettes were formed and manually isolated and dissociated into single cells. The cells were expanded in Neurobasal medium supplemented with 1× NEAA, L-Glutamine (2 mM), 1× B27 supplement, and bFGF (20 ng/mL).

### 2.3. Purification of Nestin+ Cells by Fluorescence Activated Cell Sorting (FACS) and In Vitro Differentiation

Nestin+ neural progenitor cells (NPCs) were purified using fluorescence-activated cell sorting (FACS) as described previously [26,27]. Briefly, human iPSCs were induced to differentiate into NPCs to express ZsGreen and were then harvested using Accutase and resuspended in 0.1% fetal bovine serum (FBS) in 1× phosphate buffered saline (PBS) at a concentration of 5~10 × 10^6^ cells/mL. GFP/ZsGreen+ cells were purified using a FACSAria II cell sorter system (BD) at 4 °C, at a rate of 2500 cells/second. The sorted cells were re-examined by FACS for the percentage of GFP/ZsGreen+ cells to determine purity. All experiments were done in triplicate. The purified Nestin+ NPCs were maintained and expanded in DMEM/F12 supplemented with 1× NEAA, L-Glutamine (2 mM), 1× B27 supplement, 1× N2 supplement, EGF (20 ng/mL) and bFGF (20 ng/mL), and the media was changed every other day. The cells were passaged every 6–10 days. For neuronal and astrocyte differentiation, cells were seeded in 24-well plates and continued to grow in neuronal differentiation medium (Neurobasal medium plus 1× NEAA, L-Glutamine (2 mM), 1× B27 supplement, NT3 (10 ng/mL), BDNF (10 ng/mL) and GDNF (10 ng/mL)) for 2 weeks or astrocyte differentiation medium (DMEM/F12 supplemented with 1× NEAA, L-Glutamine (2 mM), 1× B27 supplement, 1× N2 supplement, and BMP4 (40 ng/mL)) for 4 and 12 weeks, respectively. The differentiation medium was changed every 3 days. For whole-cell patch-clamp recording, the differentiated neurons (14 days after differentiation) were maintained in the oxygen saturation artificial cerebrospinal fluid (119 mm NaCl, 26.2 mm NaHCO_3_, 2.5 mm KCl, 1 mm NaH_2_PO_4_, 1.3 mm MgCl_2_ and 10 mm glucose.) The patch pipette with outer diameter contained solution for whole-cell patch-clamp recording contained 90 mM K^+^-gluconate, 40 mM KCl, 1 mM MgCl_2_, 10 mM NaCl, 10 mM EGTA (ethylene glycol tetraacetic acid), 4 mM Mg-ATP and 10 mM HEPES/KOH (pH 7.4). Differentiated neurons were hyperpolarized to −90 mV, stepped to a defined voltage as indicated, and returned to −70 mV before the next cycle was run. Spontaneous action potential-like events were recorded in current-clamp mode (0 pA).

### 2.4. In Vitro Immunocytochemistry

The undifferentiated and differentiated Nestin+ cells were characterized by immunocytochemistry [26,27]. Briefly, cells grown on glass coverslips were fixed with 2% paraformaldehyde for 10 min at 4 °C and then washed 3 times in PBS. The fixed cells were incubated in blocking buffer (PBS containing 5% goat serum, 1% bovine serum albumin, and 0.1% Triton X-100) for 30 min at room temperature (RT) and were then followed in blocking buffer containing primary antibodies at the indicated concentrations overnight at 4 °C. Appropriate secondary antibodies were used for single and double labeling. All secondary antibodies were tested for cross-reactivity and nonspecific immunoreactivity. The following primary antibodies were used: Nestin (1:200, Abcam, Waltham, MA, USA), Sox2 (1:200, R&D Systems, Minneapolis, MN, USA), SOX1 (1:200, R&D Systems), OCT 4 (1:500, Abcam), SSEA4 (1:10, Development studies hybridoma Bank, DSHB, Iowa City, IA, USA), GFAP (1:4000, DAKO/Agilent, Carpinteria, CA, USA), tubulin βIII (1:1000, Sigma, St. Louis, MO, USA), MAP2 (1:200, Sigma), human nuclei antigen (hN) (1:200, Millipore, Burlington, MA, USA), NeuN, (1:200, Sigma), and neurofilament-light chain (NFL) (1:200, Sigma). Bis-benzamide (DAPI, 1:1000; Sigma) was used to visualize the nuclei. Images were captured using a Zeiss Axiovision microscope with z-stack split view function.

### 2.5. Transplantation of Nestin-ZsGreen+ Cells into a Rat Cervical SCI Model

Since expression of ZsGreen was significantly downregulated after expansion of a few passages in vitro, the purified Nestin+ NPCs were relabeled with EGFP so the grafted cells could be tracked in vivo after being transplanted to the injured spinal cord. Briefly, lentiviral vector expressing EGFP was constructed using pLenti6.3/V5-DEST (Thermo Fisher Scientific, Waltham, MA, USA, Cat. No. V53306) as the backbone, with the EGFP expression cassette driven by a constitutively on CMV promoter. Lentiviruses produced in 293FT cells were added to Nestin+ NPCs culture, and positively labeled cells were selected with blasticidin for 7 days. After infection and selection, approximately 50% of NPCs expressed GFP. Two hours before transplantation, NPCs were detached from the dishes, collected by centrifugation at 200× *g* for 4 min, and resuspended in 1 mL culture medium. Cell count and viability was assessed with trypan blue in a hemacytometer, and then the cell suspension was centrifuged a second time and resuspended in a smaller volume to give a density of 5 × 10^4^ viable cells/μL. All animal care and surgical interventions were undertaken in strict accordance with the approval of the Animal Welfare Committee at the University of Texas Health Science Center in Houston. Surgical procedures were performed as described previously [28,29,30]. Briefly, after anesthetization with Ketamine (50–80 mg/kg BW, IP) and Xylazine (5–10 mg/kg BW, IP), 28 adult (3–6 months old, half male and half female) NIH nude rats (RNU Rat, NIH-Foxn1^rnu^, Charles River) received a dorsal laminectomy at the 5th cervical vertebral level (C5) to expose the spinal cord and then a 150 kdyne moderate unilateral contusive injury was performed using an Infinite Horizons (IH) impactor (Infinite Horizons LLC, Lexington, KY, USA) with the spine stabilized using steel stabilizers inserted under the transverse processes one vertebra above and below the injury. In addition, 4 nude rats (2 male and 2 female) received a dorsal laminectomy alone as sham-operated controls. After the contusion, the wound was sutured in layers and bacitracin ointment (Qualitest Pharmaceuticals, Huntsville, AL, USA) was applied to the wound area. Rats received 0.1 mL of gentamicin (2 mg/mL, sc, ButlerSchein, Dublin, OH, USA) and then recovered on a water-circulating heating pad. The post-surgery care also included the treatment of analgesic agent, buprenorphine (0.05 mg/kg, sc; Reckitt Benckise, Hull, UK), twice a day for 3 days and daily monitoring for 7 days. At Day 15 post-injury, rats with C5 unilateral contusion were randomly assigned to three groups, which received DMEM (injury control), human fibroblasts (cell transplantation control), and hiPSC-NPCs, respectively. There were 8 nude rats (4 male and 4 female) in each group. Rats were re-anesthetized as above and the laminectomy site was re-exposed. Three injections were made at the injury center, 2 mm cranial and caudal to the injury center, respectively, at a depth of 1.1 mm and 0.6 mm laterally from the midline. At each site, 1 μL of cell suspension was injected through a glass micropipette with an outer diameter 50~70 μL and the tip sharp-beveled to 30–50° at rate of 0.2 μL/min as described previously [26,28,30]. Thus, a total of 150,000 cells were grafted into each injured spinal cord. The animals were allowed to survive for 8 weeks after transplantation for behavioral and histological analyses. 

### 2.6. Behavioral Analyses

Rats were habituated and tested for 3 days in horizontal ladder (HL) and grooming tests prior to SCI. The HL and grooming tests were performed weekly for 10 weeks. Animals were coded and behavioral assessments and analyses were performed by two investigators blinded to the treatment groups. 

Previous studies [31,32,33] showed that the grooming test was sensitive to sensorimotor deficits after cervical SCI. The grooming test was scored 0–5 according to the highest point where the tested forelimb reached after cool tap water was applied to the animal’s head and back with soft gauze. Grooming activity was recorded with a video camera (Sony Handycam HDR-CX440, Digital HD Video Camera Recorder) from grooming onset through at least two stereotypical grooming sequences (around 2 min). Slow motion video playback was used to score each forelimb independently as described previously [33]. Score 5 was the highest with rats having the full range of motion with the forepaw being able to touch the area of the head behind the ears, and 0 the lowest with the rat not being able to contact any part of the face or head. Scores 4 to 1 were given if the animal’s forepaw contacted the front, but not the back, of the ears (4), the eyes and the area up to, but not including, the front of the ears (3), the area between the nose and the eyes, but not the eyes (2) and the underside of the chin and/or the mouth area, but not the nose (1), respectively. 

The horizontal ladder (HL) test is sensitive to locomotor deficits caused by cervical SCI as this test requires adequate sensorimotor function to feel the ladder rungs and contact the rungs during stepping [32,33,34]. The device consists of a walkway enclosed by Plexiglas walls (10 cm tall, 100 cm long, and 7.6 cm apart) with wooden applicator sticks being inserted irregularly (from 0.5 to 1.5 cm apart). The home cage was placed at the end of the walkway to encourage the animal towards the “target” end point. In addition, the red enrichment cylinders blocked off the end of the runway in the home cage to prevent the tested animal escaping. Before injury, animals were habituated to the HL apparatus in three training sessions with each session having 5–10 complete transits. The rats were trained to travel from the start point towards the goal box without turning around and, if required, gently guided by the experimenter. A testing session included two successful walks without turning or back walking. The horizontal tests were recorded on a video camera angled perpendicular to the rungs. The percentage of missed steps (# misstep/# total steps × 100) were calculated using slow motion video playback for each walk and the average of both walks was the percent misstep for each week. 

The changes of mean grooming scores and HL misstep percentages over time was analyzed using repeated measures ANOVA with the between groups factor. The differences among the groups at a survival time over the 10 post-injury testing weeks were performed using Tukey HSD post hoc *t*-tests.

### 2.7. Immunohistochemistry

After the last behavioral tests at 10 weeks post injury (PI), rats were anesthetized with a cocktail of ketamine and xylazine (80 mg/kg and 10 mag/kg, respectively, ip) and then sacrificed by transcardial perfusion with 4% paraformaldehyde (PFA) in PBS. The injured spinal cord segments were removed and follow these processes: post-fixation in 4% PFA overnight, dehydration in 20% sucrose overnight, and 30% sucrose overnight at 4° C. The cords were then embedded in OCT compound (Fisher Scientific, Waltham, MA, USA) and cryosectioned in 20 µm slices either transversely or longitudinally and mounted serially on Superfrost Plus Gold Slides. 

For immunofluorescent staining, the slides were blocked in Tris buffered saline (TBS) containing 0.2% Triton X-100 (TBST) and 10% donkey serum for 1 h at RT. The sections were then incubated overnight at 4 °C in TBST containing 5% donkey serum containing monoclonal mouse anti-human nuclei with or without polyclonal chicken anti-GFP with the following antibodies: SOX9 and GFAP (markers for astrocytes), NeuN and MAP2 (markers for neurons), OLIG2 and NG2 (markers for oligodendrocytes and oligodendrocyte precursor cells), or SOX2 and Nestin (markers for undifferentiated NPCs). After three washes of 5 min in TBS, sections were incubated in TBST containing 5% donkey serum and the appropriate fluorescence-conjugated secondary antibodies for 1 h at RT. The sections were rinsed in TBS and coverslipped with ProLong^®^ Gold antifade reagent. For the control sections, the primary antibodies were replaced with normal mouse IgG and normal rabbit IgG and then the same staining procedures including the same combination of secondary antibodies were followed. A Zeiss Observer Z1 inverted fluorescence microscope was used to capture representative images at 20× resolution. Photomicrographs were assembled using Adobe Photoshop^®^ and Adobe Illustrator^®^ software (Adobe, San Jose, CA, USA). For quantitation of neuronal or astrocyte differentiation of grafted hiPSC-derived NPCs, images from four coronal sections, 200 µm apart, were taken spanning the lesion epicenter from each animal and three random 20× fields from each section were used for quantification. Since EGFP only labeled around half the grafted NPCs, we used human nuclei antigen to calculate the total numbers of grafted NPCs in each field. The percentages of NeuN+ neurons/hN or GFAP+ astrocytes/hN were quantified using Zeiss Zean quantitative software (Carl Zeiss, Jena, Germany). For ratio analysis, unbiased stereological counting methods were not needed [35]. 

### 2.8. Quantification of Gray Matter by MAP2-IR 

Hemisection images of the injured side were captured beginning at the epicenter and every 400 µm traversing rostrally and caudally using 20× resolution on a Zeiss Observer Z1 inverted microscope. The injured hemisections were outlined from rostral 2 mm to caudal 2 mm, and a set threshold was used to automatically calculate the area of MAP2 within the selected region using customized image analysis macros in Zen software. The normalized percentage of MAP2 was found by dividing the area of MAP2 by the normalized gray matter area. The normalized gray matter area for each animal was the average of the gray matter areas of three rostral and three caudal hemi-sections, which were far away from the injury. The total volumes of spared gray matters in the injured side around the injury (4 mm and 2 mm from the epicenter caudally and rostrally, respectively) were calculated by total spared gray matter areas (from C2000 µm to R2000 µm) × distance (400 µm). 

### 2.9. Spared White Matter by EC Staining 

The spared white matter was determined as previously described [29,33,36]. Briefly, one set of slides (every 10th section, each set containing 10 serial sections spaced 200 µm apart) was stained with eriochrome cyanine staining to identify spared myelinated white matter (WM) as described previously [37]. The lesion epicenter was defined as the section containing the least amount of spared white matter. White matter sparing was defined as tissue showing normal myelin appearance and density (lacking cysts and degeneration). The spared white matter at the injury side was measured at hemi-sections every 400 µm apart for a total distance of 2000 µm rostral and caudal to the injury epicenter. The percentage of spared WM was calculated by dividing the spared WM area in each hemi-section to the total normalized WM area. The total normalized WM area in each animal was the average of WM areas at three rostral and three caudal hemi-sections, which were far away (at least 4 mm) from the injury. 

The mean MAP2-IR areas or EC-stained areas in the injured side at each distance from epicenter were tallied by injured groups. Differences in MAP2-IR or EC-stained areas between the groups across distance rostrally or caudally from the injury was assessed using a Repeated Measures ANOVA and Tukey’s HSD post hoc testing (SPSS version 28, SPSS Inc., Chicago, IL, USA), with the between subjects factor as experimental group and the within subjects factor as repeated measures on distance from the epicenter of the injury.

## 3. Results

### 3.1. Neural Progenitor Cells Derived from Nestin-ZsGreen iPSC Knockin Reporter Cell Line

Nestin-ZsGreen iPSC knockin reporter cells were first differentiated into embryoid bodies (EB), at which point some cells started to express Nestin and the fluorescence reporter ZsGreen (Figure 1A–C). A large percentage of cells (>80%) in the EBs had started to glow green by day 12 of differentiation. When these cells were enzymatically dissociated with Accutase and replated on Matrigel, in addition to continuously expressing Nestin and ZsGreen, they also formed typical neural rosette structures (Figure 1D–F), an early primary structure for neuroectoderm and NPCs. To obtain pure NPCs that expressed Nestin and ZsGreen, we sorted the ZsGreen+ cells with FACS, and the ZsGreen+ cells all expressed Nestin as well as SOX1, a specific and well-accepted NPC marker (Figure 1G–J). The purified NPCs were able to proliferate in vitro for many passages without significantly changing their core properties, such as uniform expression of SOX1 and Nestin, and potent proliferation capacity and differentiation potential. Importantly, no undifferentiated hiPSCs, identified by pluripotency markers SSEA4 (Figure 1J–L) OCT4 (Figure 1M–O), were found in the purified NPCs. After proliferation for 10 passages in vitro, the purified NPCs significantly down-regulated the expression of ZsGreen even though Nestin continued to be expressed ubiquitously (Figure 1P–R).

To examine the differentiation potential, the purified NPCs were induced to differentiate in neuronal and astrocyte differentiation medium, respectively. At 2 weeks in neuronal differentiation medium, many NPCs differentiated into mature neurons expressing MAP2 (Figure 2A) and NFL (Figure 2B). These differentiated neurons were able to generate action potentials during patch clamp recording (Figure 2C). These results showed that NPCs had matured into functional neurons. At 1 month in the astrocyte differentiation medium, many NPCs differentiated into GFAP+ astrocytes with stellate morphology and long processes (Figure 2D), which were likely immature astrocytes. At 3 months after differentiation, most astrocytes became more mature with increased size in cytoplasm and fewer processes (Figure 2E). These results indicate that NPCs derived from hiPSCs have the potential to differentiate into both neurons and astrocytes. 

### 3.2. Survival and Differentiation of Grafted iPSC-Derived Neural Progenitor Cells

NIH nude rats received moderate C5 unilateral contusion SCI (IH 150 kdyne). Two weeks after injury, hiPSC-derived ZsGreen NPCs were transplanted into the spinal cord of injured rats. At 2 months after transplantation, robust survival of grafted NPCs was found in all hosts. Most grafted cells, as labeled by lentiviruses encoding GFP, remained in areas around the injury epicenter to fill the injury cavity (Figure 3A,B). The grafted NPCs were found in the injured side without migration into the uninjured contralateral side (Figure 3A). Many grafted NPCs had neuronal morphology with processes extending into the spared host spinal cord (Figure 3C,D). To further confirm the neuronal differentiation of grafted NPCs, its expression of neuron-specific markers was examined. As shown in Figure 4A–D, around 29% (29% ± 7%) of grafted NPCs differentiated into NeuN+ neurons at the injured epicenter where the host neurons were lost. Some grafted NeuN+ neurons were found in white matters around the injury epicenter. These newly differentiated neurons were labeled by human nuclei antigen (hN), a specific human cell marker, and/or GFP (Figure 4B–E), indicating they are derived from transplanted hiPSC-derived NPCs. 

The grafted NPCs integrated well with host spinal cord around the injury epicenter (Figure 4F). While the majority of grafted NPCs remained in the injured epicenter, some were able to migrate into the host spinal cord across the astroglia scar around the injury. In addition, around 67% (67% ± 12%) of the grafted NPCs that were labeled with human specific nuclei antigen and some also with GFP was shown to express GFAP, indicating that they have differentiated toward the astrocyte lineage (Figure 4G–J). Different from the host hypertrophic reactive astrocytes in the scar border (Figure 4G–J, arrows below white line), the graft-derived astrocytes had stellate morphology with small cell bodies and relatively longer processes. Importantly, the host axons labeled by pan-NF regrew and extended into the NPC graft (Figure 4K–M).

### 3.3. Spared Gray and White Matters after Transplantation of iPSC-NPCs

To examine the effect of grafted NPCs on the survival of host neurons, we use MAP2 immunohistochemistry to quantify spared gray matter in animals who received grafts of NPCs, human fibroblasts, or culture medium, respectively, at 2 months after transplantation (Figure 5A). The statistical analyses showed that the main effect of group on spared MAP2-IR area was significant (F = 124.524, df = 2.473, 29.673, *p* < 0.001). The gray matter areas were significantly greater in rostral 0.4 (*p* = 0.0013 between NPC and fibroblast groups and between NPC and medium groups), 0.8 (*p* = 0.00079 between NPC and fibroblast groups and *p* = 0.00081 between NPC and medium groups), and 1.2 (*p* = 0.0322 between NPC and fibroblast groups and *p* = 0.0225 between NPC and medium groups) or caudal 2 mm (*p* = 0.046 between NPC group and medium group) from injury epicenter, respectively, in rats receiving NPC grafts compared to the groups receiving human fibroblasts or culture medium (Figure 5B). The total spared gray matter volume in the injury side was also significantly larger in the NPC graft group compared to the other two groups (Figure 5C). These data indicates that iPSC-NPC grafts contributed to increased spare gray matter and lessened the severity of the injury by reducing neuronal loss. We also quantified the spared white matter around the injury using EC staining (Figure 6A). The statistical analyses showed that there was a significant interaction effect between the injury and the detection site (F = 6.663, df = 8.160, 48.959, *p* = 0.000006). The spared WM in the NPC groups was significantly higher than the fibroblast group at C2000 (*p* = 0.0283) and C1600 (*p* = 0.0459) (Figure 6B). Although there was a trend of increased spared white matter areas in the NPC group compared to the groups receiving human fibroblasts or culture medium in all other distances rostrally and caudally from the injury epicenter, the differences were not statistically significant (Figure 6B).

### 3.4. Improved Forelimb Functional Recovery after Transplantation of Human iPSC-Derived NPCs

To assess whether transplantation of human iPSC-derived NPCs would improve upper limb/brachial plexus recovery of cervical SCI animals, the grooming and horizontal ladder tests, two well-accepted forelimb outcome measures, were performed. In the grooming test [31,34], scores were determined by the animals’ grooming activity recorded after cool water was applied to their head. Two independent raters were blinded and scored the recordings separately to ensure unbiased results. Animals were tested every week starting from 1 week to 10 weeks post contusion (i.e., 8 weeks post grafting). Both the injured ipsilateral side (left side, L) and the uninjured contralateral side (right side, R) were scored. Our results showed that grooming scores were significantly lower in all groups at all tested time points in the injured ipsilateral side compared to the uninjured contralateral side, in spite of the gradual recovery in the injured side from 1 week to 4 weeks after contusion (Figure 7A). These results suggest that contusion caused permanent functional deficits in the left forelimb. However, the grooming scores in the injured side were significantly higher in the iPSC-NPC group compared with both human fibroblast controls and the culture medium controls of the ipsilateral side at 8 (*p* = 0.001 between NPC and fibroblast groups and *p* = 0.003 between NPC and medium groups), 9 (*p* = 0.009 between NPC and fibroblast groups and *p* = 0.001 between NPC and medium groups) to 10 (*p* = 0.002 between NPC and fibroblast groups and *p* = 0.003 between NPC and medium groups) weeks post-injury (Figure 7A). Results from the horizontal ladder test [32,38] showed a similar trend of forelimb functional recovery in the iPSC-NPC grafted animals (Figure 7B). Compared to human fibroblast controls and the culture medium controls, the iPSC-NPC grafted group had significantly lower percentages of misstep post graft at 6 (*p* = 0.012 between NPC and fibroblast groups and *p* = 0.005 between NPC and medium groups), 8 (*p* = 0.044 between NPC and fibroblast groups and *p* = 0.021 between NPC and medium groups), 9 (*p* = 0.033 between NPC and fibroblast groups and *p* = 0.005 between NPC and medium groups) or 10 (*p* = 0.023 between NPC and medium groups) weeks after SCI. Taken together, these data show that transplantation of hiPSC-derived NPCs can promote functional recovery of the forelimb after cervical SCI. 

## 4. Discussion

In this study, we transplanted NPCs that were derived from human iPSCs to a well-established rat cervical spinal cord injury model. We tracked the differentiation and integration of the transplanted cells and evaluated the forelimb functional recovery post-grafting. Our results indicate that grafted iPSC-NPCs were able to integrate into the injured spinal cord and differentiate into neurons and glia and, importantly, significantly improve forelimb functional locomotor recovery. Our results suggest that transplantation of human iPSC-derived NPCs could be an effective therapy for treating patients after SCI.

Transplantation of iPSC-NPCs could provide several benefits for SCI repair. Purified NPCs can differentiate into glial cells and neurons in vitro and survive and maturate into astrocytes and neurons after transplantation into traumatically injured spinal cord. These results are consistent with previous studies [20,22,26], suggesting the great potential of NPC transplantation to replace lost neurons and astrocytes in the injured spinal cord. The graft-derived neurons extend their processes into host spinal cord both rostrally and caudally, suggesting a potential to serve neuronal relays reconnecting the injured descending and ascending long tracts with their denervated targets in spinal cord caudal and rostral to the injury, respectively. Previous studies showed that young neurons differentiated from grafted NPCs could overcome the inhibitory microenvironment in the injured spinal cord to extend processes into the host spinal cord for relatively long distances [17,18]. A combination of growth factors with NPC transplantation may further enhance the survival of grafted neurons and the extensive growth of their process and, importantly, promote functional recovery. These studies suggest that delivery of growth factors can further enhance the therapeutic efficacy of NPC transplantation [17,18]. Our results also showed that grafted NPCs survived and filled in the injury cavity and differentiated into astrocytes, which may form a favorable microenvironment to promote the regeneration of host axons, such as the corticospinal tract [39,40]. Grafted astrocytes may also enhance the survival of co-grafted neurons and the formation of synaptic connections to grafted neurons. Thus, the glial and neuronal differentiation from grafted hiPSC-NPCs could work synergistically to enhance the formation of neural relays reconnecting the injured descending and ascending axons after SCI. However, further studies are needed to determine how the newly formed neuronal relays may contribute to functional recovery after SCI. For example, it remains unclear which long tracts form synaptic connection with grafted neurons and how the neuronal relays with each tract may contribute to functional recovery. Combining specific anterograde tracing for each tract with immune-electronic microscopy could provide solid anatomic evidence about the formation of neural relay between grafted neurons and descending/ascending tracts. Connection mapping using trans-synaptic rabies viruses, especially the monosynaptic modified rabies viruses, could provide useful insights about the integration and functional connection of graft-derived neurons [41,42]. In addition, deactivation/activation of grafted neurons using chemo-genetic or optogenetic approaches could be very helpful to determine the contribution of grafted neurons in functional recovery [43,44]. It will be very interesting to determine whether different neuronal phenotypes differentiated from grafted NPCs could play varying roles in restoring functional recovery. These future studies could be very useful not only to understand the underlying mechanism for functional recovery but also to optimize the promising therapeutic option of hiPSC-NPCs after SCI. 

Our results also showed that transplantation of hiPSC-derived NPCs increased the spared GM and WM, suggesting that grafted NPCs play an important neuroprotective role to decrease neuronal and myelin loss after SCI. The grafted NPCs and their subsequent derivatives (e.g., glial cells) could provide neuroprotection by releasing multiple neurotrophins to enhance the survival of host neurons and oligodendrocytes and/or modifying the injury microenvironment to decrease the injury [45,46,47,48,49,50]. Effects of grafted NPCs in the spared WM were less effective than in spared GM. One potential explanation is that the cervical contusion used in this study injured GM more than WM. Another reason could be that the grafted NPCs are mainly located in the injured center, mainly around GM, with less grafted NPCs migrating in WM. It will be interesting to test whether inducing NPCs into glial precursor cells before transplantation will further enhance their glial differentiation and migration in the WM and thus potentially enhance their protection to host WM. Our results suggest that both neuroprotection and cell replacement could contribute to functional recovery by transplantation of hiPSC-derived NPCs following SCI. We observed that around 4% of grafted NPCs did not express either a mature neuron marker or astrocyte marker in the injured center. Future studies will investigate the cell types of these cells using different cell specific markers, especially the immature progenitor cell markers, such as sox2 and sox9. Previous studies showed that transplanted hiPSC-NPCs took a long time for maturation [51,52]. It will be informative to track the differentiation of grafted NPCs after long-term survival (1 to 2 years post-transplantation). These projects are ongoing. We will examine whether a longer survival time will lead to more maturation of grafted NPCs. More importantly, we will monitor whether the grafted NPCs will continue to proliferate and any potential tumorigenesis. These long-term safety studies will be critical for the successful translation of hiPSC-derived NPCs in the future. 

The phase II clinical trials using NPCs derived from fetal brains for SCI have been suspended due to limited efficacy and side effects caused by immunosuppression and transplantation [53,54]. Additionally, fetal cells vary from batch to batch, and transplantation of fetal tissues frequently causes extensive ethical concerns. All of these disadvantages for fetal brain derived cells point towards a need for alternative sources for cell-based therapy in clinics. Human iPSC-derived neural progenitor cells could serve as a better cell resource for SCI therapy. First, they could be readily produced homogenously in bulk. They are reprogrammed from somatic cells with minimal ethical controversy. In addition, they can be manufactured from the patient’s own somatic cells and function as wonderful autologous grafts, which theoretically do not require immunosuppression treatment after being grafted. However, the cost of manufacturing hiPSC for the patient’s own use and the time for deriving the NPCs or other desired cells for SCI or other neurological diseases may limit the clinical translation of hiPSCs in the future [55]. Development of “universal hiPSCs” could overcome these limitations and offer “off-the-shelf” cells derived from hiPSC for human diseases including SCI [56,57,58]. In summary, our results show that transplantation of hiPSC-derived NPCs promotes functional recovery of the forelimb after cervical SCI. Both neuronal and glial replacement and neuroprotection contribute to functional recovery. Our results suggest that transplantation of hiPSC-derived NPCs is a potentially effective therapy for patients after SCI. Long-term safety and efficacy studies will be critical for the translation of hiPSC-derived NPCs in the future. Development of “off-the-shelf” universal hiPSC will further speed up the translation. 

## Figures and Tables

**Figure 1 cells-11-02765-f001:**
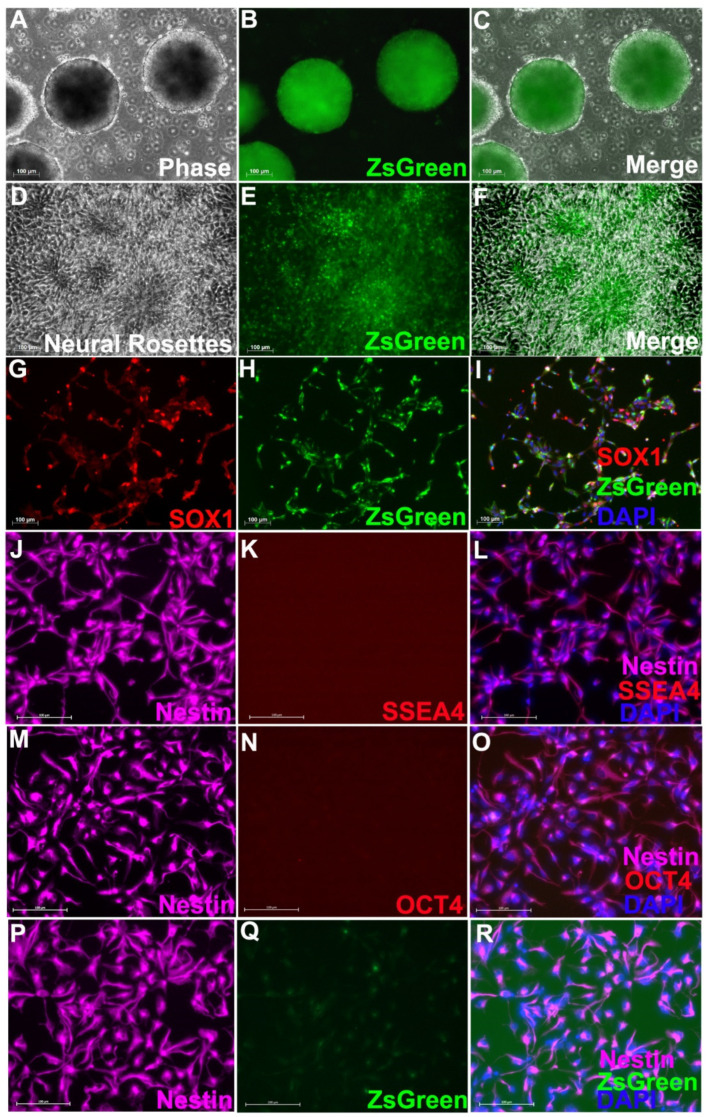
**Neural differentiation of Nestin-ZsGreen hiPSC knockin cell line.** After 12 days of neural induction as embryoid bodies (EB), cells started to express Nestin-ZsGreen (**A**–**C**). When dissociated and replated, they formed a typical neural rosette structure, while continuing to express ZsGreen (**D**–**F**). After FACS for ZsGreen, almost all GFP+ cells expressed SOX1, a well-accepted and specific neural stem cell marker (**G**–**I**). The purified NPCs do not contain undifferentiated iPSCs as evidenced by lack of expression of SSEA4 (**J**–**L**) and OCT4 (**M**–**O**), two commonly used pluripotency markers. Before being grafted to animals, expression of ZsGreen (**P**–**R**) was significantly downregulated although Nestin continued to be expressed ubiquitously. Scale Bar: 100 mm.

**Figure 2 cells-11-02765-f002:**
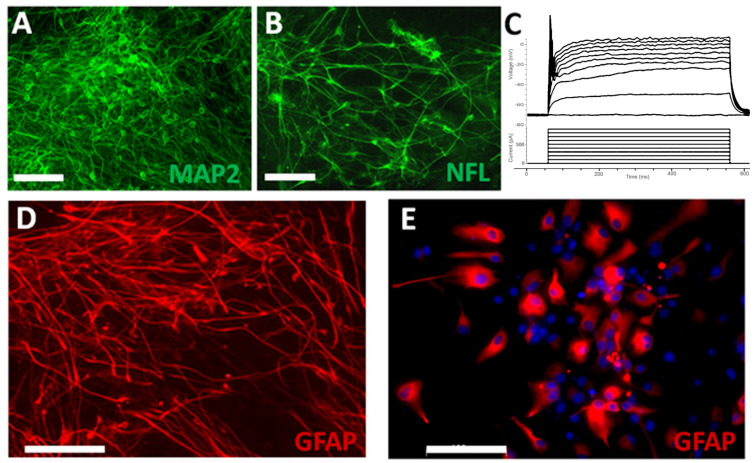
**Differentiation of the purified hiPSC-derived NPCs.** After differentiation in the neuronal differentiation medium for 2 weeks, many NPCs differentiated into mature neurons expressing MAP2 (**A**) and NFL (**B**). These differentiated neurons were able to generate action potentials during patch clamp recording (**C**). After differentiation in the astrocyte differentiation medium for 1 month, many NPCs differentiated into immature GFAP+ astrocytes with stellate morphology and long processes (**D**). After differentiation for 3 months, most astrocytes became more mature with increased size in cytoplasm and fewer processes (**E**). Scale bars: 100 μm.

**Figure 3 cells-11-02765-f003:**
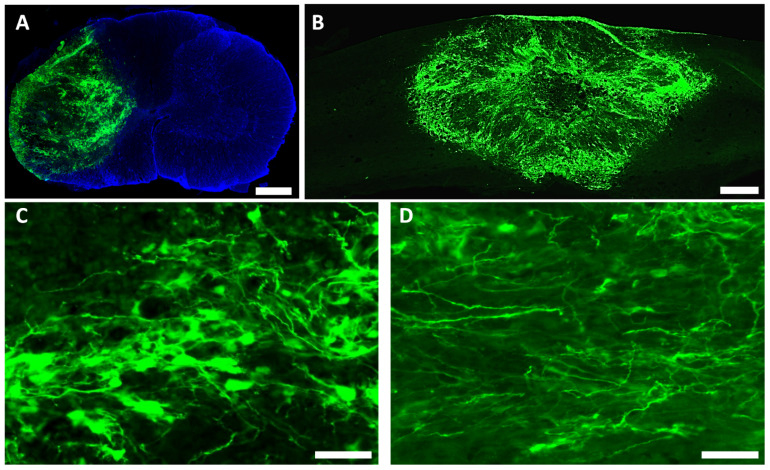
**Survival of grafted hiPSC-derived neural progenitor cells.** Human iPSC-derived NPCs were transplanted at 2 weeks after moderate C5 unilateral contusion SCI (IH 150 kdyne) in nude rats. Robust survival of grafted NPCs was observed in all animals that received hiPSC-NPC grafts at 2 months after transplantation (**A**,**B**). The grafted NPCs filled the lesion cavity and primarily remained in areas around the injury epicenter at the injury side (**A**,**B**). Grafted hiPSC-NPCs differentiated into cells with the morphology of neurons (**C**), with long processes extending to the spared host spinal cord (**D**). Scale bars: (**A**,**B**), 500 μm; (**C**,**D**), 50 μm.

**Figure 4 cells-11-02765-f004:**
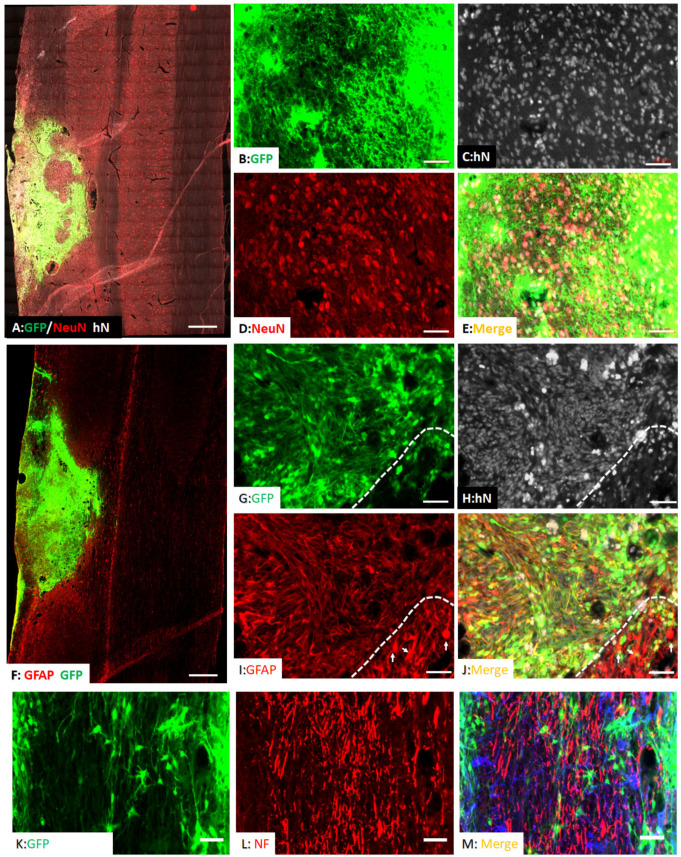
**Differentiation of grafted hiPSC-derived NPCs after SCI.** At 2 months after transplantation, 29% of grafted hiPSC-NPCs differentiated into mature NeuN+ neurons in the injured epicenter, where the host neurons were lost (**A**). In high magnification (**B**–**E**), NeuN+ neurons in the injured areas also expressed GFP and human nuclei (hN), indicating these neurons are derived from grafted hiPSC-NPCs. 67% of grafted NPCs differentiated into GFAP+ astrocytes in the injured spinal cord at 2 months after transplantation (**F**–**J**). Different from the host hypertrophic reactive astrocytes in the scar border (**I**,**J**, arrows, white lines indicate the edge of glial scar), the graft-derived astrocytes had stellate morphology with small cell bodies and relatively longer processes. The host axons integrated well with the grafted NPCs and extended into the graft (**K**–**M**). Scale bars: (**A**,**F**), 500 µm; (**B**–**E**,**G**–**J**,**K**–**M**), 50 µm.

**Figure 5 cells-11-02765-f005:**
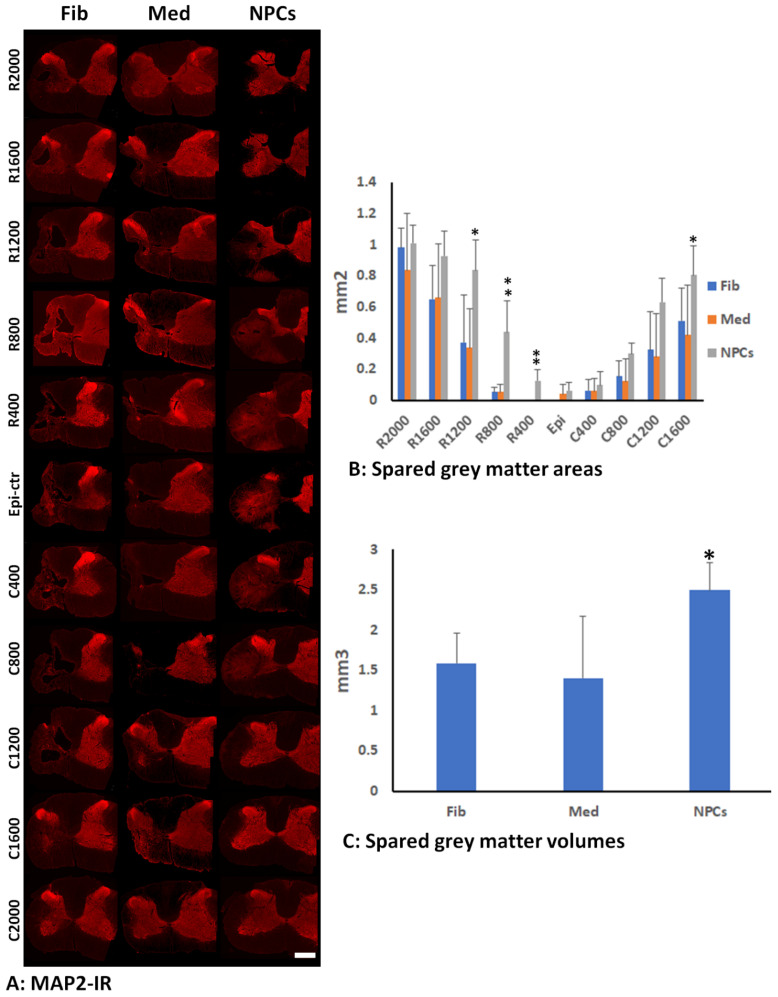
**Spared gray matter after transplantation of hiPSC-NPC.** Spared gray matter (GM) in the injured side was quantified using MAP2 immunohistochemistry at 2 months after transplantation (**A**). The spared GM areas were significantly greater in rostral 0.4, 0.8, and 1.2, or caudal 1.6 mm from injury epicenter, respectively, in rats receiving NPC grafts compared to groups receiving human fibroblasts or culture medium (**B**). The total spared GM volume in the injury side was also significantly greater in NPC grafting group compared to other two groups (**C**). Scale bar: A, 500 μm. Data in (**B**,**C**) represent the man ± SD (n = 5). * in (**B**,**C**), *p* < 0.05; ** in (**B**), *p* < 0.01.

**Figure 6 cells-11-02765-f006:**
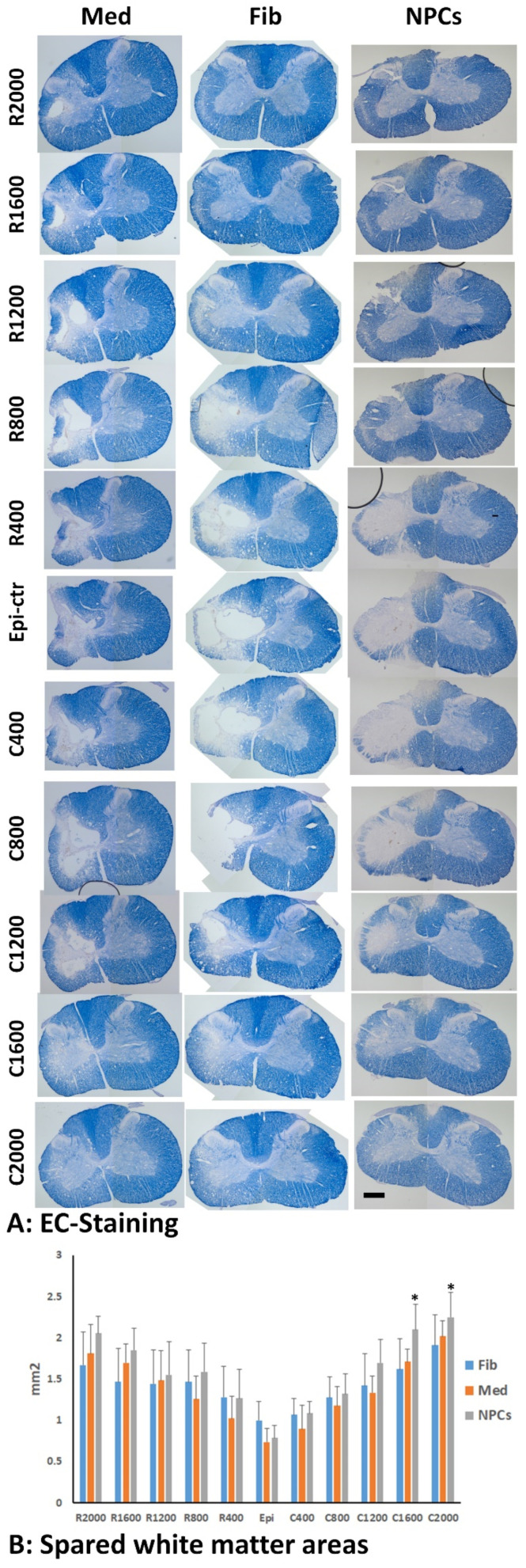
**Spared white matter after transplantation of hiPSC-NPC.** Spared white matter (WM) in the injured side was quantified using EC-staining at 2 months after transplantation (**A**). The spared WM areas were significantly greater in caudal 1.6 and 2 mm from injury epicenter in rats receiving NPC grafts compared to groups receiving human fibroblasts (**B**). Scale bar: (**A**), 500 μm. Data in (**B**) represent the man ± SD (n = 5), * *p* < 0.05.

**Figure 7 cells-11-02765-f007:**
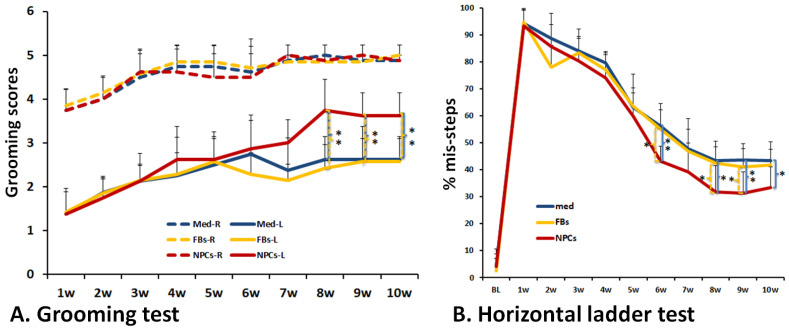
**Functional recovery after transplantation of hiPSC-derived NPCs.** Human iPSC-derived NPCs were transplanted at 2 weeks after moderate C5 unilateral contusion SCI (IH 150 kdyne) in nude rats. Functional recovery of the forelimb was assessed by the grooming and horizontal ladder tests. Grooming scores (**A**) in the injured side (solid lines) were significantly greater in animals receiving NPC grafts compared to group receiving human fibroblasts or culture medium, respectively, starting 8 weeks after injury (6 weeks after transplantation). The percentages of misstep in the injured forelimb, as measured by the horizontal ladder test, were also significantly lower in the NPC grafting group compared to the other two groups at 6, 8, 9, and 10 weeks after injury (**B**). Data in (**A**,**B**) represent the man ± SD (n = 8), * in (**B**), *p* < 0.05; ** in (**A**,**B**), *p* < 0.01.

## Data Availability

Data are available from the author with the permission of Florida International University and University of Texas Health Science Center at Houston.

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
