# Peer review of "Transplantation of Human Induced Pluripotent Stem Cell-Derived Neural Progenitor Cells Promotes Forelimb Functional Recovery after Cervical Spinal Cord Injury"

_cells, 2022, doi:10.3390/cells11172765_

Round 1

Reviewer 1 Report

The study has an intriguing and pertinent idea and showed methodological care in only some areas, however, it was unclear how well the results could be reproduced and what sample size was employed. It has significant qualities for a pilot study and may be valid for guiding future studies, according to the results reported, but the statements made might be less forceful.

The method section of the manuscript did not mention about control group. There was any control group for the surgery model or treatment proposal?  Amount of animals used in the studies? 

Results - In the legend of Figure 2 there are two descriptions for the A and B parts, which is correct? In Figure 3 legend have a description for M-P items, but the figure stopped in the M item. In Figure 4 item C, the bar of the NPC group is described as ips

The use of human fibroblast or saline as a comparison condition did not mention in the method description.  

What was the proposal to use human fibroblasts as one of the controls?

In the behavior results, is it not clear what the comparison performed for the significant results shown in Figure 6?

The discussion showed many affirmations, but the results are not clear regarding the reproducibility of the method.

Author Response

We would like to thank the reviewer for his/her comprehensive and constructive comments/suggestions, which we believe have significantly improved our revised manuscript. Please find the point-to-point detail for the revisions we made in this revised version manuscript.  

The study has an intriguing and pertinent idea and showed methodological care in only some areas, however, it was unclear how well the results could be reproduced and what sample size was employed. It has significant qualities for a pilot study and may be valid for guiding future studies, according to the results reported, but the statements made might be less forceful.

The method section of the manuscript did not mention about control group. There was any control group for the surgery model or treatment proposal?  Amount of animals used in the studies? 

Answers: The three groups (injured control, cell transplantation control with human fibroblasts, the NPC transplantation) and the numbers of animal in each group are now included in the revised manuscript.

Results - In the legend of Figure 2 there are two descriptions for the A and B parts, which is correct? In Figure 3 legend have a description for M-P items, but the figure stopped in the M item. In Figure 4 item C, the bar of the NPC group is described as ips

Answers: The previous figure 2 is now figure 3. The two descriptions for A and B parts are both correct with one describing the survival of grafted NPCs and another for the location and migration. The previous figure 3 and 4 are now the figure 4 and 5, respectively. The mislabeling and typo in both figures are now corrected.

The use of human fibroblast or saline as a comparison condition did not mention in the method description.  

Answers: The use of human fibroblasts or medium (DMEM) is now added to methods.

What was the proposal to use human fibroblasts as one of the controls?

Answers: human fibroblasts was used as cell transplantation control.

In the behavior results, is it not clear what the comparison performed for the significant results shown in Figure 6?

Answers: The significant results are among NPC group against injured control group (Medium) or cell transplantation control group (Fibroblasts), respectively, in the injured side of forelimb. The details are now added to results and figure legends.

The discussion showed many affirmations, but the results are not clear regarding the reproducibility of the method.

Answers: The detailed methods and results are now added to the revised manuscript. These details will increase the reproducibility.

Reviewer 2 Report

In the article “Transplantation of human induced pluripotent stem cell-derived neural progenitor cells promotes forelimb functional recovery after cervical spinal cord injury”, the researchers demonstrated grafted iPSC-NPCs can integrate into the injured spinal cord and differentiate into neurons and glia, and improve forelimb functional locomotor recovery. Currently, it has been demonstrated transplantation of hiPSC-derived NPCs can promote functional recovery after SCI, but most studies have primarily used thoracic SCI models focusing on the hindlimb functional recovery. However, most clinical SCI cases occur at the cervical level, and recovery of hand and forearm function is one of the highest priorities  to  individuals  with  SCI. The motivation behind the problem investigated in this manuscript is interesting and meaningful, however, several issues still need to be addressed.

1.    Fig1: The number of Fig 1 legend is confusing. Fig 1 K doesn’t show MAP2. Fig 1 JNFM is misspelled. Fig 1 H doesn’t show name of GFP.

2.    Fig 1: The magnification of Fig 1 L-M should be consistent to prove that with differentiation, the mature astrocytes are morphologically distinguished from immature astrocytes.

3.    In results “Importantly, no undifferentiated hiPSCs, identified by pluripotency markers OCT4 and SSEA4, were found in the purified NPCs (data not shown).” Means NPC all differentiate in vitro. However, in discussion “Some of these grafted NPCs expressed SOX2 and/or SOX9, suggesting they are immature precursor cells.” Means some NPCs are immature precursor cells. It should be discussed why NPCs are differentiated in vitro but undifferentiated in vivo.

4.    Fig 1: Why the authors labeled neurons with NFL but not NeuN or MAP2, which are commonly used as neuronal markers?

5.    Fig 2 A lacks scale bars.

6.    Fig 2 D: How can it be demonstrated that “Many grafted NPCs had neuronal morphology with long processes extending into host spinal cord both caudally and rostrally”?

7.    Fig 3: The number of Fig 3 legend is confusing.

8.    Fig 3: The figures of the transplanted NPC differentiated into NeuN and GFAP were not arranged in the same order.

9.    Fig 3: Why the human nucleistaining expressed by NeuN+ neurons is gray but human nucleistaining expressed by GFAP+ astrocytesis blue?

10.  Fig 4 does not show representative images of C1600, C2000, R1600, and R2000. Due to there are statistical differences between groups at C2000, representative images of C1600, C2000, R1600, and R2000 should be shown.

11.  Fig 5 does not show representative images of C1600, C2000, R1600, and R2000. Due to there are statistical differences between groups at C1600 and C2000, representative images of C1600, C2000, R1600, and R2000 should be shown.

12.  In results of Fig 5, “The statistical analyses showed that there was a significant interaction effect between the distance from the injury and the experimental group.”, there is an errorinthe expressionof“between the distance from the injury and the experimental group”. It should be “between the injury and detection site”?

13.  In results of Fig 6, “These results suggest that that contusion caused permanent functional deficits in the left forelimb”. There are two “that”,one of them should be deleted.

14.  In discussion, “Our results also showed that transplantation of hiPSC-derived NPCs increased the spared gray matter (GM) and WM, suggesting that grafted NPCs play important neuro-protective role to decrease neuronal and oligodendrocyte loss after SCI.” However, oligodendrocytes were not mentioned in the results of this study.

15.  In discussion, “Both neuronal and glial replacement and neuroprotection contribute to functional recovery”, where a space is missing between “to” and “functional”.

16.  In discussion, “A combination of growth factors with NPC transplantation may further enhance the survival of grafted neurons and the extensive growth of their process, and importantly, promote functional recovery. These studies suggest that delivery of growth factors can further enhance the therapeutic efficacy of NPC transplantation.” Growth factors are not mentionedin the results of this study, so references should be added.

17.  In discussion “Some of these grafted NPCs expressed SOX2 and/or SOX9, suggesting they are immature precursor cells.” However, the results presented in this manuscriptdo not mention that the grafted NPCs express SOX2 and/or SOX9.

Author Response

We would like to thank the reviewer for his/her comprehensive and constructive comments/suggestions, which we believe have significantly improved our revised manuscript. Please find the point-to-point detail for the revisions we made in this revised version manuscript.

In the article “Transplantation of human induced pluripotent stem cell-derived neural progenitor cells promotes forelimb functional recovery after cervical spinal cord injury”, the researchers demonstrated grafted iPSC-NPCs can integrate into the injured spinal cord and differentiate into neurons and glia, and improve forelimb functional locomotor recovery. Currently, it has been demonstrated transplantation of hiPSC-derived NPCs can promote functional recovery after SCI, but most studies have primarily used thoracic SCI models focusing on the hindlimb functional recovery. However, most clinical SCI cases occur at the cervical level, and recovery of hand and forearm function is one of the highest priorities  to  individuals  with  SCI. The motivation behind the problem investigated in this manuscript is interesting and meaningful, however, several issues still need to be addressed.

  1. Fig1: The number of Fig 1 legend is confusing. Fig 1 K doesn’t show MAP2. Fig 1 JNFM is misspelled. Fig 1 H doesn’t show name of GFP.

Response: The mislabeling in figure 1 is now corrected.

  1. Fig 1: The magnification of Fig 1 L-M should be consistent to prove that with differentiation, the mature astrocytes are morphologically distinguished from immature astrocytes.

Responses: The previous figure 1L-M is now the figure 2. The magnifications for immature astrocytes (figure 2D) and mature astrocytes (figure 2E) are now the same.

  1. In results “Importantly, no undifferentiated hiPSCs, identified by pluripotency markers OCT4 and SSEA4, were found in the purified NPCs (data not shown).” Means NPC all differentiate in vitro. However, in discussion “Some of these grafted NPCs expressed SOX2 and/or SOX9, suggesting they are immature precursor cells.” Means some NPCs are immature precursor cells. It should be discussed why NPCs are differentiated in vitro but undifferentiated in vivo.

Responses: Oct4 and SSEA4 are markers for undifferentiated hiPSCs. We have consistently showed that there is no undifferentiated hiPSCs in the purified NPCs in vitro (IHC) and in vivo with out teratoma formation. Expression of both sox2 and sox9 identified subpopulation(s) glial precursor cells. We will quantified the grafted NPCs without both NeuN and GFAP in the future. This is now discussed.

  1. Fig 1: Why the authors labeled neurons with NFL but not NeuN or MAP2, which are commonly used as neuronal markers?

Responses: MAP2 is now added to figure 2A. NFL is considered a more mature neuronal markers than both NeuN and MAP2.

  1. Fig 2 A lacks scale bars.

Response: The previous figure 2A is now the figure 3A. The scale bar is now added.

  1. Fig 2 D: How can it be demonstrated that “Many grafted NPCs had neuronal morphology with long processes extending into host spinal cord both caudally and rostrally”?

Responses: The previous figure 2D is now figure 3D. The processes were extended both caudally and rostrally.

  1. Fig 3: The number of Fig 3 legend is confusing.

Responses: The previous fig3 is now fig 4. The mislabeling and typo in this figure are now corrected.

  1. Fig 3: The figures of the transplanted NPC differentiated into NeuN and GFAP were not arranged in the same order.

Responses: The previous fig3 is now fig 4. The differentiation of grafted NPCs into NeuN (Fig. 4B-E) and GFAP (Fig. 4G-J) are now arranged in the same order.

  1. Fig 3: Why the human nucleistaining expressed by NeuN+ neurons is gray but human nucleistaining expressed by GFAP+ astrocytesis blue?

Responses: The previous fig3 is now fig 4. The staining for human nuclei is now the same (gray).

  1. Fig 4 does not show representative images of C1600, C2000, R1600, and R2000. Due to there are statistical differences between groups at C2000, representative images of C1600, C2000, R1600, and R2000 should be shown.

Responses: The previous fig4 is now fig 5. The representative images of C1600, C2000 R1600 and R2000 are now added.

  1. Fig 5 does not show representative images of C1600, C2000, R1600, and R2000. Due to there are statistical differences between groups at C1600 and C2000, representative images of C1600, C2000, R1600, and R2000 should be shown.

Responses: The previous fig5 is now fig 6. The representative images of C1600, C2000 R1600 and R2000 are now added.

  1. In results of Fig 5, “The statistical analyses showed that there was a significant interaction effect between the distance from the injury and the experimental group.”, there is an errorinthe expressionof“between the distance from the injury and the experimental group”. It should be “between the injury and detection site”?

Response: The typo is now corrected.

  1. In results of Fig 6, “These results suggest that that contusion caused permanent functional deficits in the left forelimb”. There are two “that”,one of them should be deleted.

Responses: The previous fig6 is now fig 7. The typo “that” is now deleted.

  1. In discussion, “Our results also showed that transplantation of hiPSC-derived NPCs increased the spared gray matter (GM) and WM, suggesting that grafted NPCs play important neuro-protective role to decrease neuronal and oligodendrocyte loss after SCI.” However, oligodendrocytes were not mentioned in the results of this study.

Responses: EC staining shows the spared myelin. Now it is corrected that “grafted NPCs play important neuro-protective role to decrease neuronal and myelin loss after SCI.

  1. In discussion, “Both neuronal and glial replacement and neuroprotection contribute to functional recovery”, where a space is missing between “to” and “functional”.

Response: The “to” is now added.

  1. In discussion, “A combination of growth factors with NPC transplantation may further enhance the survival of grafted neurons and the extensive growth of their process, and importantly, promote functional recovery. These studies suggest that delivery of growth factors can further enhance the therapeutic efficacy of NPC transplantation.” Growth factors are not mentionedin the results of this study, so references should be added.

Response: The citations are now added.

  1. In discussion “Some of these grafted NPCs expressed SOX2 and/or SOX9, suggesting they are immature precursor cells.” However, the results presented in this manuscriptdo not mention that the grafted NPCs express SOX2 and/or SOX9.

Response: This part in discussion has been revised accordingly.

Reviewer 3 Report

Spinal cord injury (SCI) is serious neurological conditions and results in motor disability in patient. Currently, effective treatment ways are not available and is ultimately required to improve patient’s quality of life. Zheng and his/her colleagues performed transplantation of iPSCs-differentiated NPCs into rat model and assessed their usefulness by analyzing 1) behavior of transplanted cells and  2) improvement of locomotor function. Although several transplantation experiments have been implemented so far, most of pre-clinical study focused on the hindlimb function recovery in thoracic SCI. Zheng et al. established a rat contusion model at cervical level 5 of spinal cord and assessed upper limb movement. Although their study is important for development of future stem cell therapy, several critical technical concerns remain in the manuscript.

Major concerns:

1. Authors infected GFP-expressing lentivirus in ZsGreen-sorted iPSC-derived NPCs, due to silencing of the reporter fluorescence after passaging. However, GFP and ZsGreen are not distinguishable, and this experimental design causes many issues in subsequent imaging analysis (See following comments). Since Nestin is early NPC marker and silencing of Nestin-ZsGreen is important evidence of the differentiation and maturation of the transplanted cells. I think that the best design of the experiment is to use non-overlapped fluorescent colors.

In addition, authors should describe the detail of GFP-expressing lentiviral vector (e.g. backbone, Is this from AddGene, if so, add AddGene ID? etc).

2. There is also a risk that a small portion of ZsGreen- cells (maybe undifferentiated hiPSCs) is expanded with their high proliferation rate, and the proportion of ZsGreen+ cells (NPCs) is decreased during experiments. Although authors described that “no differentiated hiPSCs were found in the purified NPCs (data not shown)”, this is very important information in their experimental design. Authors must explain how to quantify OCT4/SSEA4 (immunostaining, qPCR etc) and provide the data.

3. Although authors described “To obtain pure NPCs that expressed Nestin and ZsGreen, we sorted the ZsGreen+ cells with FACS, and the ZsGreen+ cells all expressed Nestin as well as SOX1, a specific and well-accepted NPC marker (Fig. 1G-J)”, authors have never measured Nestin expression directly. Nestin expression may be estimated by green fluorescent color, but it is unclear whether the green color is derived from Nestin-ZsGreen or lentiviral GFP.  In addition, “Nestin-GFP” is unclear, since nestin reporter is ZsGreen, but not GFP in Fig. 1.

4. Although authors described “long processes extending to the spared host spinal cord”, it is unclear whether the transplanted cells retained in the injured area or were extended. At least, it is difficult to interpret the extension from Fig. 2D.

5. I think that Fig. 3 is one of the most important results to show efficiency of the transplantation experiment. However, descriptions related to Fig. 3 are very vague like “many”, “a portion” or “some”…. The most serious risk of stem cell transplantation is that the grafted cells continue to proliferate and become tumor. Thus, it is very important how many grafted cells (or percentage) are differentiated into neuron or astrocyte and retain as immature precursors. To clearly characterize the efficiency of the transplantation, I strongly recommend the quantification (e.g. percentage of GFP+/NeuN+…) by using imaging software.

Minor concerns:

1. Authors should describe how to measure action potential in method (related to Fig. 1K).

2. Authors should describe what error bar represents (Fig. 4B, 4C, 5B, and 6).

3. Is statistical test one-sided or two-sided? In bar plots, error bar is so large and it is unclear whether the difference is really statistically significant… Addition of data points into barplot is also helpful to verify whether the difference is convincing or not.

Author Response

We would like to thank the reviewer for his/her comprehensive and constructive comments/suggestions, which we believe have significantly improved our revised manuscript. Please find the point-to-point detail for the revisions we made in this revised version manuscript.

Spinal cord injury (SCI) is serious neurological conditions and results in motor disability in patient. Currently, effective treatment ways are not available and is ultimately required to improve patient’s quality of life. Zheng and his/her colleagues performed transplantation of iPSCs-differentiated NPCs into rat model and assessed their usefulness by analyzing 1) behavior of transplanted cells and  2) improvement of locomotor function. Although several transplantation experiments have been implemented so far, most of pre-clinical study focused on the hindlimb function recovery in thoracic SCI. Zheng et al. established a rat contusion model at cervical level 5 of spinal cord and assessed upper limb movement. Although their study is important for development of future stem cell therapy, several critical technical concerns remain in the manuscript.

Major concerns:

  1. Authors infected GFP-expressing lentivirus in ZsGreen-sorted iPSC-derived NPCs, due to silencing of the reporter fluorescence after passaging. However, GFP and ZsGreen are not distinguishable, and this experimental design causes many issues in subsequent imaging analysis (See following comments). Since Nestin is early NPC marker and silencing of Nestin-ZsGreen is important evidence of the differentiation and maturation of the transplanted cells. I think that the best design of the experiment is to use non-overlapped fluorescent colors.

Responses: We partially agree with this reviewer that labeling the grafted NPCs with a different fluorescent epitope will be a better option. Unfortunately, expression of ZsGreen was significantly downregulated although Nestin continued to be expressed ubiquitously (Figure 1P-R). These results indicate the ZsGreen is not reliable marker for Nestin expression or the differentiation/maturation in grafted NPCs. We labeled the grafted NPSs with EGFP to track the survival and morphology of grafted NPCs. Since EGFP was not expressed by all grafted NPCs, we used the double staining of hN and NeuN or GFAP for the neuronal or astrocyte differentiation of grafted NPCs. So the labeling the grafted NPCs with EGFP will not change our major finding and conclusions.

In addition, authors should describe the detail of GFP-expressing lentiviral vector (e.g. backbone, Is this from AddGene, if so, add AddGene ID? etc).

Responses: the detail of EGFP-expressing lentiviral vector is now included.

  1. There is also a risk that a small portion of ZsGreen- cells (maybe undifferentiated hiPSCs) is expanded with their high proliferation rate, and the proportion of ZsGreen+ cells (NPCs) is decreased during experiments. Although authors described that “no differentiated hiPSCs were found in the purified NPCs (data not shown)”, this is very important information in their experimental design. Authors must explain how to quantify OCT4/SSEA4 (immunostaining, qPCR etc) and provide the data.

Responses: We used immunohistochemistry to quantify the expression of OCT4 and SSEA4. We provided the data in figure 1 J-L and M-O.

  1. Although authors described “To obtain pure NPCs that expressed Nestin and ZsGreen, we sorted the ZsGreen+ cells with FACS, and the ZsGreen+ cells all expressed Nestin as well as SOX1, a specific and well-accepted NPC marker (Fig. 1G-J)”, authors have never measured Nestin expression directly. Nestin expression may be estimated by green fluorescent color, but it is unclear whether the green color is derived from Nestin-ZsGreen or lentiviral GFP.  In addition, “Nestin-GFP” is unclear, since nestin reporter is ZsGreen, but not GFP in Fig. 1.

Responses: We used immunohistochemistry to directly examine Nestin expression in Figure 1J, M, P, respectively.  The typo Nestin-GFP is now corrected to nestin-ZsGreen.

  1. Although authors described “long processes extending to the spared host spinal cord”, it is unclear whether the transplanted cells retained in the injured area or were extended. At least, it is difficult to interpret the extension from Fig. 2D.

Responses: The previous figure 2 is now figure 3. Figure 3B showed most grafted NPCs remained in the injured cavity. We agreed with this reviewer that more studies are needed to confirm whether these extended processes form the synaptic connection with host neurons. We have discussed some future experiments which are needed to interpret the potential role of these process in neuronal relay and functional recovery after SCI.  

  1. I think that Fig. 3 is one of the most important results to show efficiency of the transplantation experiment. However, descriptions related to Fig. 3 are very vague like “many”, “a portion” or “some”…. The most serious risk of stem cell transplantation is that the grafted cells continue to proliferate and become tumor. Thus, it is very important how many grafted cells (or percentage) are differentiated into neuron or astrocyte and retain as immature precursors. To clearly characterize the efficiency of the transplantation, I strongly recommend the quantification (e.g. percentage of GFP+/NeuN+…) by using imaging software.

Responses: We have quantified the percentages of neurons (NeuN/hN) and astrocytes (GFAP/hN) using immunohistochemistry. The data has now added to results and figure legend (the previous figure 3 is now figure 4).

Minor concerns:

  1. Authors should describe how to measure action potential in method (related to Fig. 1K).

Response: the method for measure action potential is now added.

  1. Authors should describe what error bar represents (Fig. 4B, 4C, 5B, and 6).

Responses: The error bars in these figures were SD. This information is now added to figure legends.

  1. Is statistical test one-sided or two-sided? In bar plots, error bar is so large and it is unclear whether the difference is really statistically significant… Addition of data points into barplot is also helpful to verify whether the difference is convincing or not.

Responses: As described in methods and results, the statistical analyses for behavioral tests were performed in the injured side only. We carefully confirmed the differences in grooming and horizontal tests in figure 7 were statistically significant. We have now added the p values at weeks 6 to 10 post-injury between NPC group and injury control (Medium) and cell transplantation control (Fibroblasts) group, respectively.

Round 2

Reviewer 1 Report

There are only a few corrections left to make:

The term "Briefly" is repeated at line 136.

-24 or 28 animals in total were used? Please correct this information. Line 150 stated that there were 24 adult NIH naked rats, but line 156 stated that there were 4 sham rats and 8 naked rats in each of the three groups.

-At line 166, the phrase "three injections" makes it appear as though three injections were administered to the same animal, but I believe the writers meant to refer to the three different types of injections (DMEN, human fibroblast, and hiPSC-NPC). It ought to be "The injections," as I recommend.

Author Response

The term "Briefly" is repeated at line 136.

Response: The Typo is now deleted.

-24 or 28 animals in total were used? Please correct this information. Line 150 stated that there were 24 adult NIH naked rats, but line 156 stated that there were 4 sham rats and 8 naked rats in each of the three groups.

Response: A total of 28 rude rats (24 for injury and 4 for sham) were used. 

-At line 166, the phrase "three injections" makes it appear as though three injections were administered to the same animal, but I believe the writers meant to refer to the three different types of injections (DMEN, human fibroblast, and hiPSC-NPC). It ought to be "The injections," as I recommend.

Response: Actually, three injections were made for each injured animal at epicenter, 2 mm cranial and 2 mm caudal to epicenter, respectively. The description in our revised manuscript is correct. 

Reviewer 3 Report

Authors addressed my concerns well, and the revised manuscript is significantly improved. I still have one comment, but I believe that this manuscript is almost ready for publication.

Related to my major concern 4:

Line 332: "Many grafted NPCs had neuronal morphology with long processes extending into host spinal cord both caudally and rostrally (Fig. 3C-D)."

Although it is clear that the transplanted hiPSC-NPCs showed neuronal morphology, it is difficult to detect caudal and rostral extension in Fig 3C-D. Thus, please 1) indicate the direction of caudal and rostral extension (e.g. adding arrows), 2) replace them with clearer figures showing caudal and rostral extension, or 3) revise the description.

Author Response

Related to my major concern 4:

Line 332: "Many grafted NPCs had neuronal morphology with long processes extending into host spinal cord both caudally and rostrally (Fig. 3C-D)."

Although it is clear that the transplanted hiPSC-NPCs showed neuronal morphology, it is difficult to detect caudal and rostral extension in Fig 3C-D. Thus, please 1) indicate the direction of caudal and rostral extension (e.g. adding arrows), 2) replace them with clearer figures showing caudal and rostral extension, or 3) revise the description.

Response: The description is revised as "Many grafted NPCs had neuronal morphology with processes extending into the spared host spinal cord (Fig. 3C-D)".